



# Monitoring abiotic and biotic parameters of forest regrowth under different management regimes on former wildfire sites in northeastern Germany – data from the PYROPHOB project

Marie-Therese Schmehl[1], Yojana Adhikari[2], Cathrina Balthasar[3], Anja Binder[2], Danica Clerc[4], Sophia Dobkowitz[1], Werner Gerwin[5], Kristin Günther[6], Heinrich Hartong[7], Thilo Heinken[8], Carsten Hess[9], Pierre L. Ibisch[2], Florent Jouy[2], Loretta Leinen[9], Thomas Raab[10], Frank Repmann[10], Susanne Rönnefarth[2], Lilly Rohlfs[4], Marina Schirrmacher[4], Jens Schröder[4], Maren Schüle[8,2], Andrea Vieth-Hillebrand[6], and Till Francke[1]

[1]Institute of Environmental Science and Geography, University of Potsdam, Karl-Liebknecht-Straße 24–25, 14476 Potsdam, Germany
[2]Centre for Econics and Ecosystem Management, Eberswalde University for Sustainable Development, Alfred-Möller-Str. 1, 16225 Eberswalde, Germany
[3]Senckenberg German Entomological Institute, Eberswalder Str. 90, 15374 Müncheberg, Germany
[4]Landeskompetenzzentrum Forst Eberswalde, Landesbetrieb Forst Brandenburg, Alfred-Möller-Str. 1, 16225 Eberswalde, Germany
[5]Research Center Landscape Development and Mining Landscapes (FZLB), Brandenburg University of Technology Cottbus-Senftenberg, Siemens-Halske-Ring 8, 03046 Cottbus, Germany
[6]GFZ Helmholtz Centre for Geosciences, 14473, Potsdam, Germany
[7]Büro UmLand Schmid und Hartong GbR, Berkenbrücker Dorfstr. 11, 14947 Nuthe-Urstromtal, Germany
[8]General Botany, Institute for Biology and Biochemistry, University of Potsdam, Maulbeerallee 3, 14469 Potsdam, Germany
[9]Naturwald Akademie gGmbH, Roeckstraße 40, 23568 Lübeck, Germany
[10]Chair of Geopedology and Landscape Development, Brandenburg University of Technology Cottbus-Senftenberg, Siemens-Halske-Ring 8, 03046 Cottbus, Germany

**Correspondence:** Marie-Therese Schmehl (schmehl@uni-potsdam.de)

**Abstract.** We present the data recorded by eight institutions within the PYROPHOB project, running from 2020 to 2024 at two forest research sites in the south-west of Brandenburg, Germany. The aim of the project was to monitor abiotic and biotic parameters of forest regrowth under different management regimes on former wildfire sites in northeastern Germany. The observations comprised intermittent and continuous measurements or surveys on meteorological parameters (rainfall, temperature, vapour pressure deficit), soil (soil type and texture, soil chemistry and leaching, water content, soil temperature), deadwood, stand structure, vegetation regrowth, abundance of selected fauna (moths, beetles, mammals), UAV-based remote sensing, and photo monitoring. Thus, the multitude of collected data allows not only for detailed analyses of these observables separately, but also considering their interaction for a more multidisciplinary view on forest recovery after a wildfire. The data are available under the following DOIs: 10.23728/b2share.5821b57eae3b45619c2263205b1c9815 (Part 1, (Schmehl et al., 2025a)), under embargo unitl 2026-03-31 (Part 2, (Schmehl et al., 2025b)), 10.23728/b2share.d49631857d1345c48be9e36b62214e6d (Part 3.1, (Schmehl et al., 2025c)), and 10.23728/b2share.51bdf4b6dc854873b6ff44fdddbf4c3b (Part 3.2, (Schmehl et al., 2025d))



# 1 Introduction

## 1.1 Background and context

Forests provide numerous ecosystem services essential for natural systems and society. However, they have experienced increasing stress over recent years, mainly due to climate change, natural disasters, shifting species composition, invasive species, and wildfires (Ayres et al., 2014). Consequently, managing forests in a way that improves their resilience against these hazards is a major challenge. Adapted management comprises not only the time period before the above-mentioned hazards arise, but also after their occurrence for restoring the damaged ecosystems and enhancing their future stability. Because of the related long-term impact in forest systems, any of the related management decisions are of particular importance.

Among the above-mentioned hazards, wildfires present one of the most fundamental threats, as they potentially obliterate biological materials entirely. A high-intensity wildfire causes substantial economic losses and severely impairs the ecosystem services of the affected area. The recognition of the risk associated with wildfires and the importance of managing affected forests has grown in recent years, especially with regard to increased pressure on forest functions due to climate change (Stevens-Rumann et al., 2018). However, there is still a considerable shortage in datasets looking at post-fire forest development (Zang et al., 2024). Forest ecosystems comprise an intricate spectrum of relevant processes and interactions across multiple disciplines (e.g. forestry, botany, entomology, hydrology, soil science, etc.). However, research on forest recovery tends to focus on specific topics such as soil (e.g. Naethe et al., 2018; Klose and Makeschin, 2005), nutrients (e.g. Li et al., 2024), stand structure (e.g. Tesha et al., 2024), soil and vegetation (e.g. Gustafsson et al., 2021), soil microbiology (e.g. Yang et al., 2024), insects (e.g. Schauermann, 1980), or climate conditions (e.g. Wolf et al., 2021), which hinders a holistic assessment of the relevant processes. More importantly, the associated data are not always publicly accessible to allow for reproducible or follow-up research. Notable exceptions include, e.g., the work of Zang et al. (2024), having assembled a comprehensive dataset for burnt sites based on remote sensing techniques. While such remote sensing-based datasets excel at their spatial extent and potentially global coverage, they are, obviously, restricted to properties obtainable from spectral imagery, which excludes most of the above-mentioned specific topics as they require onsite surveys.

Specifically for temperate regions, post-wildfire related studies sharing their data are scarce (Jackisch et al., 2023, as an exception). To our best knowledge, no comprehensive, multidisciplinary dataset covering different management options is available.

On the other hand, sophisticated long-term monitoring for forest sites does exist (e.g. the long-term monitoring ICP forest sites, see Michel, 2022). While such dedicated observations provide invaluable information on forest development in the long term, they are evidently not designed for an immediate onset of monitoring following an unplanned catastrophic event such as a wild fire.

PYROPHOB (project duration: 2021 - 2025) is an interdisciplinary research project aiming to fill this gap. It focuses on studying the effects of forest fires and subsequent management strategies in pine forests in Brandenburg, north-east Germany. While fires in pine stands cause high tree mortality and alter soil properties, the knowledge at ecosystem-level about their effects is limited. The project is targeted at understanding how different components of biodiversity interact in post-fire ecosystems and



how management strategies affect these interactions. In Central Europe, post-fire forest areas are typically managed through interventions like salvage logging and soil preparation. PYROPHOB explicitly considers a wider selection of management options to assess their impact on natural regeneration and other related environmental variables.

In this data paper, we present the broad range of data collected during the project. The paper is linked to an accompanying repository that ensures public access to these data and their long-term storage. In that context, this paper is intended to give a comprehensive overview of the shared data, and provide them in a consistent way. It also offers an overview of relevant other data sources and the scientific research done on the data so far. It does not – and cannot – summarize the scientific findings of the entire project.

## 1.2 Study goals and study design

The project investigates ecosystem components and their interactions in burned study sites, focusing on the effects of different forest management treatments on post-fire development. Key research questions include how forest fires change abiotic and biotic factors, how management treatments affect these factors, and how quickly new forests can be developed to be less vulnerable to fire. These different treatments were observed on dedicated "sites", each representing a specific management practice. On each site, the project uses a system of standardized sample plots to allow for spatially consistent analyses across
environmental factors, taxa, and management types.

## 1.3 Structure of this paper

This paper presents data from the majority of the research activities conducted within the PYROPHOB project between June 2020 and April 2025. Eight involved research institutions contributed to this dataset. However, some topics (i.e. fungi, predatory soil arthropods) are not included in this collection. We selected the data included in this selection mainly guided by their
uniqueness, scientific interest, and potential value for subsequent analyses and re-use.

The study area, plot design and specific methodology have been described in high detail in Heinken et al. (2024). Therefore, we limit the information here to a minimum needed to understand the published data structure. Likewise, various dedicated scientific studies have already been conducted on the data, which are summarized in Sect. 5 and which cannot be covered here. Consequently, this paper is intended as a data paper, exclusively focusing on the presentation of the data.

The study area is introduced in Sect. 2.1; the acquisition and, partly, the processing of the included subsets of the data are documented in Sect. 3. Other potentially relevant data from third parties are listed in Sect. 4. Sect. 5 provides a brief overview of the already published studies related to the presented data. The paper closes with Sect. 6 outlining research perspectives with regard to the published data set.





## 2 Study area and design

### 2.1 Study area


The PYROPHOB project was carried out in two large burned Scots pine forest areas in Brandenburg, northeastern Germany, about 50 km southwest of Berlin (see Fig. 1). Both areas are spatially distinct, unconnected (approx. 10 km apart), and are further referred to as 'Treuenbrietzen' (TB) and 'Jüterbog' (JB), named after the towns nearby.

The region is characterized by dry, sandy soils and a climate between atlantic and continental, with the vegetation mainly
consisting of Scots pine, various herb species, and dense bryophyte layers.

Treuenbrietzen (TB) experienced a forest fire in late August 2018, which destroyed 334 ha of pine stands. After the fire, different silvicultural treatments were applied, including salvage logging, planting various tree species, and leaving certain parts untreated. Some areas were fenced off to protect them from wildlife damage. In mid June 2022, a second fire occurred in TB, which affected nearly the whole formerly burnt area south of the dividing national road, and affected approximately
180 ha.

The second study area, near Jüterbog (JB), is located on a former military training ground converted into a wilderness area managed by the Brandenburg Wilderness Foundation. The foundation focuses on wilderness development and natural processes, corresponding to minimal intervention. JB was affected by a large wildfire in June 2019, which burned 744 ha of pine and mixed pioneer forest. In line with the wilderness concept, no silvicultural treatments or hunting were conducted after
the fire.

Both areas had been affected by prior military use. Apart from being suspected of repeatedly causing wildfires, the remaining ordnance and contamination constituted a significant hazard. Consequently, considerable surveying and clearing operations had to be implemented before the start of the fieldwork activities, and some adjustments to the site and plot layout had to be made.

Figure 1 provides an overview of the study areas and the designated investigation sites and plots described in the following
section.

### 2.2 Study design

#### 2.2.1 Selection of Study Sites and Plots

In 2020, two years after the fire in TB and one year after the fire in JB, 15 study *sites* were established across both *areas* (10 in TB and 5 in JB). These sites were chosen to represent a variety of forest management practices, fire intensities, and
neighbouring unburned Scots pine stands (i.e. reference sites, one in each area). The sites were selected to maximize within-site homogeneity with unit sizes around 5 ha for ensuring representativeness. The selection process involved a pre-screening using satellite imagery, geological and soil maps, digital terrain models, forest inventories, and input from local foresters. The final selection and demarcation of sites were conducted in collaboration with landowners, foresters, and forest authorities after on-site inspections under consideration of practical constraints.





**Figure 1.** Overview of the to study areas Treuenbrietzen (TB) and Jüterbog (JB) with the location of the sites with different forest management, and the included plots, marking the points of measurement.





The 13 sites in burned areas encompassed seven forest management variants, resulting from different combinations of salvage logging, ploughing, raking, planting, sowing, and fencing, with varying treatment timings. Six sites, with no forestry interventions, were included, with different fire severity, levels of game browsing, and previous stand ages. Sites were labeled with capital letters. Due to the second fire in TB destroying also the reference site G, a reference site L was added as an additional unburned site in 2022. Tab. 1 provides an aggregated overview of the attributes of each site, including pre- and post-fire

conditions, and basic treatments. A more detailed overview can be found in Heinken et al. (2024).

**Table 1.** Overview of site attributes and their basic treatments. The age of the burned pine stands was determined using forestry data and checked in the field by counting growth rings on tree stumps if possible (C: Central, E: East, N: North, S: South, W: West). Partial removal means that eather 50% (0.5) or 75% (0.75) of standing trees were removed. Aggregated from Heinken et al. (2024).

| Site | Area | Size (ha) | Forest Age (years) in 2018 | Tree Removal | Soil Treatment | Fire Year |
|------|------|-----------|----------------------------|--------------|----------------|-----------|
| B | TB | 4.9 | 64 | no removal | none | 2018 |
| C | TB | 3.2 | 71 | partial removal (0.5) | none | 2018 |
| D | TB | 3.2 | N: 68, S: 102 | partial removal (0.5) | none | 2018 |
| E | TB | 3.6 | 68 | partial removal (0.75) | raking | 2018 |
| F | TB | 10.0 | 68 | partial removal (0.75) | ploughing | 2018 |
| G | TB | 5.2 | 71 | unburned | none | unburned |
| H | TB | 3.1 | N: 45, CS: 39 | clearcut | ploughing | 2018 |
| I | TB | 3.0 | W: 98, E: 73 | clearcut | none | 2018 |
| J | TB | 5.7 | W: 46, E: 41 | clearcut | ploughing | 2018 |
| K | TB | 5.4 | SW: 64, NE: 70 | no removal | none | 2018 |
| L | TB | 2.8 | 66 | unburned | none | unburned |
| U | JB | 5.2 | N: 96, S: 76 | no removal | none | 2019 |
| V | JB | 3.0 | N: 76, S: 66 | no removal | none | 2019 |
| X | JB | 4.0 | 99 | no removal | none | 2019 |
| Y | JB | 3.1 | 29 | no removal | none | 2019 |
| Z | JB | 2.8 | 66 | unburned | none | unburned |

    Within each study site, ten *sample plots* were selected as replicates. Among these, three plots were designated as VIP ("Very Important Plots"), where most parameters were recorded. Additional plots for soil and hydrological site characterization were set up at one VIP per site. Less resource-intensive parameters were assessed across all ten plots or only the remaining seven plots, depending on the spatial variation. These remaining plots are referred to as "Individual Tree Plots" (ITP), the term having

been derived from the rejuvenation survey, as trees were marked with ID-labels there.

    In total, the project included 150 plots, comprising 45 VIP main plots and 105 ITP plots.





### 2.2.2 Data acquisition at Plot, Site and Area level

Figure 2 shows a schematic overview of the experimental setup and instrumentation on a VIP plot, a VIP soil plot, and measurements in the vicinity of the VIP plots.

At the center of each sample plot, a wooden pole with 1.5 meters height marked its exact location, which served as the reference point for data collection in the surrounding area.

    Data collection included microclimatic measurements, mapping of stand structure, deadwood volumes and quality, tree regeneration, and understory vegetation across all plots. Fungi and predatory soil arthropods (e.g., ground beetles and spiders) were recorded on a subset of plots.

Soil investigations took place just outside the boundaries of the VIPs to minimize impact. For each site, on one VIP soil plot, a soil profile was temporarily opened to characterize soil type and enable the instrumentation for measuring soil moisture, soil, and seepage water.

    In addition to these measurements associated with the plots, further data were acquired on the site level: Saproxylic beetles and moths were trapped, mammals were monitored near VIPs, UAV flights and breeding bird surveys were conducted across 130  the sites.

    Rainfall measurements took place at three locations for each of the two research areas.

## 3 Methods and data

This section describes the relevant methods used in the acquisition of the data in this data publication. More details and a complete overview of all conducted measurements are contained in Heinken et al. (2024).

### 3.1 Overview and data formats

Table 2: Overview of section 3: Summary of each data subset, main observed variables and units, temporal coverage and path in the data structure. Specific details can be found in the subsections and accompanying json-files in the repository (see also Tab. 3).

| Sect. | Data subset | Main observation variables (units) | Spatial coverage | Temporal coverage (month/year) |
|---|---|---|---|---|
| 3.2 | Geo data | extent of sites, location of plots, fire extent | both study areas | - |





| 3.3.1 | Meteo data | leaf wetness, barometric pressure, photosynthetically active radiation, total solar radiation, wind speed and direction, precipitation, air temperature at 1.3 m (°C) relative humidity | 1 station per study area | 05/21-12/24 |
|---|---|---|---|---|
| 3.3.2 | Precipitation | precipitation (mm) | 3 stations per study area | 06/20-07/24 |
| 3.3.3 | Microclimate | temperature (°C), relative humidity (%) | all / VIP plots | 04/21-01/25 |
| 3.4.1 | Humus and soil properties | bulk density (g/cm³), pH (-), electrical conductivity ($\mu$S/cm), total carbon content (%), concentrations of N, C, Ca, Fe, Mg, K, P, Al, organics, magnetic susceptibility (m³/kg) | VIP soil | 09/20-05/21, 04/24 |
| 3.4.2 | Litter decomposition | litter mass loss (g), decomposition constants | VIP plots | 05/21-10/24 |
| 3.4.3 | N mineralization | $NH_4^+$ and $NO_3^-$ contents (mg/kg) | VIP plots | 05/21-10/24 |
| 3.4.4 | Soil solution | pH, electric conductivity, cations, anions, DOC, TIC, DON | VIP plots | 10/20-10/24 |
| 3.4.5 | Soil respiration | $CO_2$-concentration (ppm) and flux (g m$^{-2}$h$^{-1}$) | VIP plots | 22-24 |
| 3.4.6 | Soil moisture and soil temperature time series | permittivity (-), soil moisture (m³/m³), soil temperature (°C) | VIP soil | 10/20-4/25 |
| 3.4.6 | Soil moisture, areal sampling | Sampling campaigns of the upper 30 cm soil moisture (m³/m³) | sites | 10/21-04/25 |
| 3.5.1 | Deadwood inventory | standing and downed deadwood (species, dimensions, degree of decomposition, position | VIP / all plots | 20-24 |
| 3.5.2 | Stand structure | number of standing trees (-), total stem volume (m³), basal area (m²), tree mean height (m), total crown projection area (m²) | VIP plots | 21-23 |
| 3.6.1 | Ground vegetation (grid) | species abundance (-) | plots | 20 |
| 3.6.2 | Ground vegetation (plots ) | species abundance (-) | plots | 20-24 |
| 3.6.3 | Tree rejuvenation | species, height (cm), root collar diameter (mm), origin, leading shoot damage, leaf damage (VIP only), status | all plots | 20-24 |
| 3.6.4 | Canopy cover | fraction cover (%) | plots | 20, 21 |



| 3.7.1 | Xylobiontic Beetles | species captured in interception and funnel traps | plots | 21-23 |
| 3.7.2 | Moths | Species captured | sites | 21-23 |
| 3.7.3 | Mammals | observations from camera traps | sites | 22-23 |
| 3.8.1 | UAV imagery | RGB-images, DEM | multi-sites | 19-23 |
| 3.8.2 | Burn indices | spatial grids of burn indices | areas | xx |
| 3.8.3 | Photo monitoring | photos | single plot per site | 21-24 |
| 3.9 | Synthesis data | aggregated synthesis data | sites | 21 |

The overall data set is organized following the concept of Fersch et al. (2020) along the observed variables and instruments. Each subset of data is documented in a dedicated metadata file in "json" format to facilitate automated meta-data retrieval. The presented data largely consist of time series or intermittent observations that were obtained at well-defined locations. These locations are documented in a geospatial data set in the format of ESRI shapefiles further described in Sect. 3.2.

The observed time series are provided in tab-separated text files in which the first column contains the date and time (in UTC, ISO 8601 format). Any other columns represent measured or derived variables. Exceptions from this data model (e.g. vegetation, manual soil measurements) are described in the subsections of this paper and documented by json files with metadata. Further details of the data repository are given in Sect. 7.

### 3.2 Geodata

The exact locations of each plot with the additional soil and climate stations, and the boundary and extent of each site are stored in two ESRI shapefiles in ("plots" and "sites", see Sect. 2.2) WGS 84 reference system. The center of each sample plot was measured with different methods and corresponding accuracy: The initial position had only been recorded via handheld GPS. In early 2023 after the second fire, all unaffectd plots were re-surveyed with a RTK dGPS. The position of the burnt plots in sites C - F were reconstructed from the orthomosaics, as the area remained closed-off. The file states the survey method and
acquisition date in the attribute table.

### 3.3 Climate

#### 3.3.1 Weather station measurements

Two meteorological stations were established — one at TB inside site H and another near the former JB military training area. Each station consists of a mast, approximately 2 m in height, equipped with seven sensors that operate continuously.
These sensors monitor leaf wetness, light intensity (including photosynthetically active radiation, PAR), barometric pressure, precipitation, solar radiation via a Silicon Pyranometer, temperature, relative humidity, and wind speed. The data was sampled at an interval of 10 seconds and stored locally every 10 minutes.

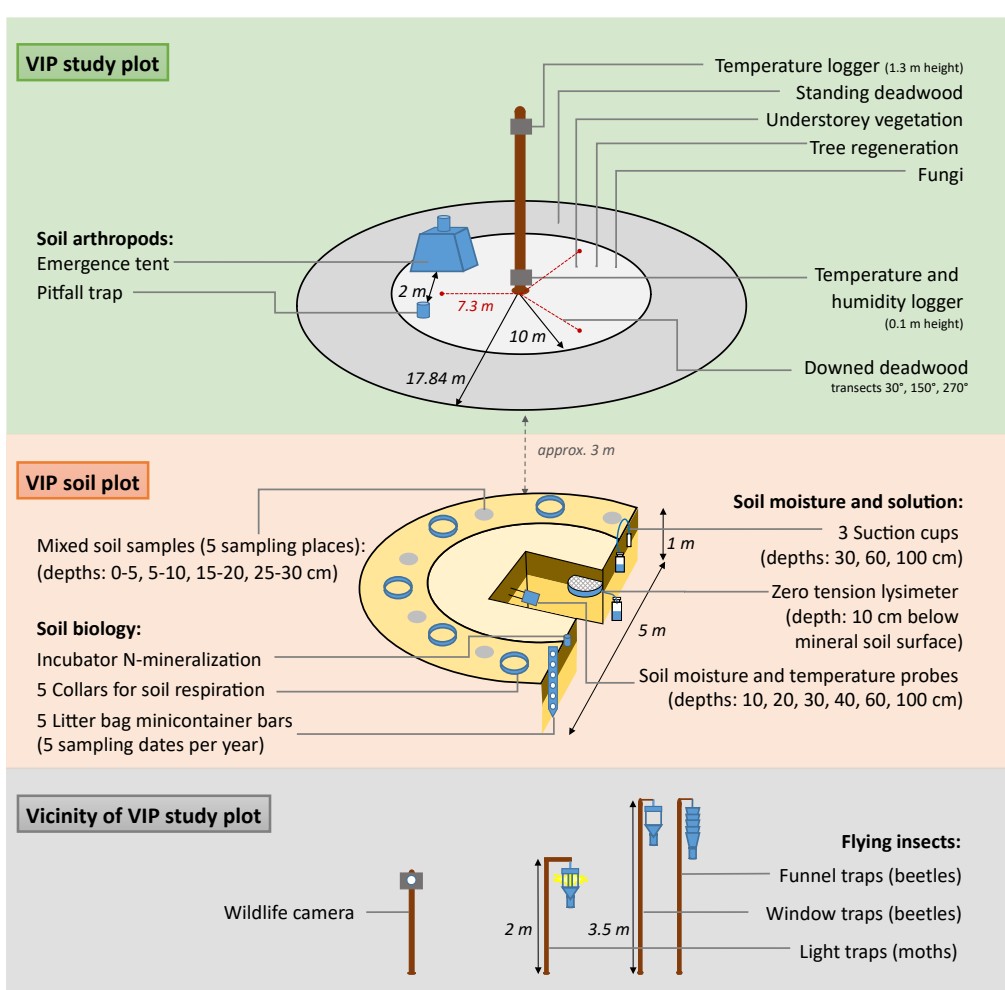

**Figure 2.** Experimental setup and instrumentation on a VIP plot, a VIP soil plot, and measurements in the vicinity of the VIP plots. Reproduced with permission from Heinken et al. (2024)





### 3.3.2 Precipitation

Additionally, in JB and TB, precipitation was measured at three locations within each study area or a maximum of 500 meters
from the nearest study site using a tipping bucket rain gauge (Davis, UK). Periods with missing or faulty rainfall data due to
battery failure, theft, or clogged instruments were removed, and then a corrected average rainfall time series was calculated for
each study area. Calibration measurements revealed an underestimation of rainfall by 15–25%, so the measured rainfall was
adjusted by +15%. To validate the measurements, the nearest rainfall station from the DWD (Felgentreu Station) was used,
which is located 2.6 and 7.2 km from the areas JB and TB, respectively. The dataset contains raw measurements and corrected
average data.

### 3.3.3 Microclimate

At each study site, air temperature was recorded using HOBO Pendant temperature data loggers (Onset, USA), which were
mounted at a height of 1.3 m in the center of each plot. To shield the sensors from direct solar radiation, the loggers were
enclosed in white plastic radiation shields. In addition, temperature and relative humidity were measured using HOBO Pro V2
sensors (Onset, USA) installed 10 cm above the ground on the three VIPs at each site. These data were used to calculate vapor
pressure deficit (VPD). VPD (kPa) was calculated as $VPD = e_s - e_a$ where $e_s$ is the saturation vapor pressure (kPa), and $e_a$
is the actual vapor pressure (kPa). Saturation vapor pressure was estimated using the equation

$$e_s = 0.6108 \cdot e^{\frac{17.27 \times T}{T+237.3}} \tag{1}$$

and actual vapor pressure was calculated as

$$e_a = e_s \cdot \frac{RH}{100} \tag{2}$$

where $T$ is the air temperature (°C) and $RH$ is the relative humidity (%), both measured at 10 cm above ground. All loggers
were synchronized to record data at 10-minute intervals. Data were stored locally and manually retrieved every 4 to 8 weeks.

### 3.4 Soil

### 3.4.1 Humus and soil properties

For investigating soil conditions after fire and after post-fire treatments, soil samples were taken in each soil sampling plot. The
initial sampling campaign took place between fall 2020 and spring 2021. The sampling campaign was repeated on most soil
plots in spring 2024 to identify possible temporal changes. Around the center of the individual sampling plots, five points were
defined for taking sub-samples from different depths. If there was a layer of litter and humus material, it was collected from an
area of 30x30 cm at each subsampling point. Soil samples were taken at these points from the mineral soil at a depth of 0-5,
5-10, 15-20, and 25-30 cm. Subsamples from one sampling plot were combined to form mixed samples per depth and plot. The
humus material was air dried (40°C) and shred ( cutting mill Pulverisette 25, Fritsch GmbH, Germany, with sieve 4 mm) and
ground (Planetary ball mill PM-400, Retsch, Germany) for analysis. The mineral soil samples were air dried, sieved (<2 mm)





and partly ground for analysis in the laboratory. For determining the bulk density of the mineral soil, soil cores of 4 cm height and a volume of 100 cm$^3$ were used.

Organic carbon (OC) content of the litter and humus material was analyzed using an elemental analyzer (CNS Analyser Vario Max Cube, Elementar, Germany). OC stock of litter and humus layers was calculated. OC content of the mineral soil samples was analyzed using an elemental analyzer. OC stocks of the mineral topsoil could be calculated by means of bulk density values determined for the mineral topsoil. For bulk density (BD) determination, the volumetric soil samples were dried at 105°C. The dry mass was weighed and related to the sample volume of 100 cm$^3$.

Electric conductivity (EC) and pH were measured in water extracts with a ratio of 1:2.5 (soil : water) using a multiparameter meter (inoLab IDS, Multi 9420, WTW, Germany, with pH- and EC probes: SENTIX 980 and TetraCon 925, both WTW). For determining effective cation exchange capacity (CECeff) and base saturation (BS) the samples for the two depths were merged to samples for the depth of 0-10 cm. The extraction method with 1 mol/L NH$_4$Cl solution developed by Lüer and Böhmer (2000) was applied, and the element concentrations (Na, K, Ca, Mg, Mn, Fe, Al) were measured by ICP-OES (iCAP 6000,

Thermo Scientific, Life Technologies GmbH, Germany). Exchangeable H$^+$ was calculated based on the pH difference in the NH$_4$Cl solution during the extraction according to Deutschland (2009). The share of base cations (Na, K, Ca, Mg) in the total extracted quantity of cations was calculated as percentage base saturation (BS).

### 3.4.2 Litter decomposition

Litter decomposition as part of carbon cycling processes was investigated using the minicontainer method introduced by Eisen-
beis et al. (1999). The minicontainers replace traditional litterbags and were filled with air dried and lightly crushed birch and pine litter (Fritsch cutting mill, with sieve 4 mm). About 200 mg of this litter was weighed in (fine scale Sartorius with 0.001 g display accuracy, Sartorius AG, Germany). The minicontainers were covered by gauze with a mesh size of 50 $\mu$m that allows mainly the soil microflora (bacteria and fungi) to access the litter material. From 2022 on, one additional container covered by a mesh with 100 $\mu$m was inserted at a depth of 2-4 cm. The minicontainer bars were vertically installed into the soil. The
installation into the soil took place in spring of each year (2021-2024). Five minicontainer bars were used in parallel at each soil sampling plot of selected study sites. The individual bars were removed during the vegetation periods in 2021 until 2024 after 2, 4, 7, 16, and 25 weeks, respectively.

After removal of the minicontainer bars from the soil, the individual litter bags were air dried and the litter mass loss determined by weighing. Decomposition rates were calculated by regression analysis using the decay function

$$m_t = m_0 \cdot e^{k \cdot t} \tag{3}$$

with $m_t$ : mass at time $t$; $m_0$ : mass at time $t = 0$; $k$ : decay constant.

### 3.4.3 Nitrogen mineralization

Net nitrogen mineralization was measured between 2021 and 2024 using the method described by Kwak et al. (2016). During the vegetation period (May to November), in situ soil core incubation was conducted for a duration of one month utilizing



stainless steel tubes, each with a diameter of 4 cm and a length of 12 cm. The upper part of the tubes was covered by metal caps to prevent input from outside. Samples taken in the field were immediately stored in a cool place and kept deep-frozen until analysis.

The inorganic N ($NH_4^+ + NO_3^-$) concentrations in the soil samples taken at the beginning of the respective incubation period as well as the concentrations after the incubation period were analyzed using FIA (see sect. 3.4.4). The inorganic N content after the incubation was compared with the respective initial inorganic N content at the beginning of the incubation. The difference between the inorganic N content of the pre-incubation and the incubation samples was calculated as net N mineralization related to the duration of the incubation period (in days).

### 3.4.4 Soil solution

The effect of fire and post-fire management on the composition of the soil solution was investigated with free-draining lysimeters and suction cups to collect solution samples from different depths.

The small **lysimeters** were made from waste water pipes and filled with glass beads to optimize their mechanical stability. They were installed 10 cm deep below the surface of the mineral soil. During installation, disturbances of the upper layers were avoided. Lysimeters were centrally installed in the soil sampling plot and solution samples were collected whenever sufficient solution was available. In the first year (2021), samples were taken monthly; in the following years, the samples were typically collected every two months during the growing season. Electric conductivity was measured by a conductivity cell (WTW, Germany) and pH was measured potentiometrically (WTW, Germany). Elemental concentrations (Na, K, Ca, Mg, P) were determined using ICP-OES (Thermo Fisher) and inorganic N ($NH_4^+$ and $NO_3^-$) with FIA (flow injection analysis, Medizin- und Labortechnik Engineering GmbH, Germany). Furthermore, dissolved organic carbon (DOC) was analyzed using a TOC-Analyzer (Shimadzu, Japan).

The **suction cups** (S12, Umwelt-Geräte-Technik GmbH (UGT), Germany) were installed on each Soil VIP plot, three vertically from the surface into depths of 30, 60 and 100 cm respectively. By applying a negative pressure of -0.5 bar using a mobile pump (MVF, UGT, Germany), the water from soil pores was sucked into the suction cups and then into separate glass bottles stored in a buried plastic box. The soil solution samples were collected typically every 2 months from November 2020 until November 2024 during the vegetation period if a sufficient volume (>10 ml) was available. Upon collection, the soil solution samples were filtered (MN619, G¼, 150 mm) within 24 hours and afterwards stored frozen until analysis. The samples with low volume (<10 ml) were diluted and all results were corrected accordingly. Soil solution analysis involved the following devices: pH (-, WTW); conductivity ($\mu$S/cm, Inlab 731 ISM, Mettler Toledo); ammonium, nitrate, phosphate concentration (mg/L, Rundküvettentest Nanocolor, Macherey Nagel, Germany); absorbance at 350 nm (-, Nanocolor Vis 2 photometer, MN); Ca, Mg, Na, K concentration (mg/L, ICP-OES iCAP 6000, Thermo Scientific, Life Technologies GmbH, Germany); DOC (mg/ml), hydrophobics (%), biopolymers (%), humics + building blocks (perc.), low molecular weights (%), humics aromaticity ($l\,mg^{-1}m^{-1}$), humics molecular weight ($g\,mol^{-1}$) and humics (%) using a Liquid chromatography – organic carbon detection (LC-OCD) system model 8 (Dr. Huber, Germany) and the chromatograms obtained were interpreted using DOC-Labor ChromCALC software program.





### 3.4.5 Soil respiration

Soil Respiration ($R_s$) was measured throughout the vegetation period (from March to November) roughly every two weeks in the years 2022, 2023, and 2024. On each Soil VIP plot, five PVC collars (75 mm height x 100 mm diameter) were inserted permanently into the soil to ca. 5 cm depth (with ca. 2-3 cm of the collar protruding above the soil) in March 2022 (March 2023 for site L) as described in Heinken et al. (2024). The height of each collar protruding above soil was measured yearly. Rs was measured using a portable infrared gas analyser (EGM-5) with a SRC-2 soil respiration chamber (150 mm height x 100 mm

diameter) (both from PP-systems). Additionally, soil temperature ($T_{soil}$, °C) and soil moisture ($M_{soil}$, % VWC) at a depth of 5 cm were measured locally for each $R_s$ measurement (HydraProbe, Stevens). The $R_s$ measurements would end after either 140 s (DT) or a change in $CO_2$ concentration of 100 ppm (DC). $R_s$ was automatically calculated by the EGM-5 program (SRL, g $CO_2$ m$^{-2}$ h$^{-1}$). The Rs values (SRL) at the end of each measurement were selected and compiled here.

### 3.4.6 Soil moisture and temperature

Soil moisture and temperature were monitored with one station at each study site, located at the soil VIP plots together with the soil chemical instruments. The measurements were recorded every 20 minutes at two replicate profiles with six different depths (10, 20, 30, 40, 60, 100 cm) using dielectric sensors (SMT-100, Truebner, Germany). In addition, two extra profiles were set up at sites F and H to compare the soil moisture dynamics between the top and furrow of the plough structures. At study site J in TB, three profile probes with nine measurement depths (SoilVUE, Campbell Scientific, UK) were installed to study intra-plot

variability.

As the study areas exhibited a high degree of small-scale variability in soil moisture, measurement campaigns were conducted using portable Frequency Domain Reflectometry soil moisture sensors (ML2x, DeltaT, UK). These campaigns covered 18 points per study area, with depth-specific sampling carried out in 5-cm increments (0-35 cm). The repeated measurements on site I allowed the selection of the most suitable structure of a linear model (simple linear model with a fixed intercept of

0) for correcting the continuous station measurements to the spatial mean. Consequently, the regression coefficient could be estimated for each site. This analysis was performed separately for the measurement depths of 10, 20, and 30 cm.

## 3.5 Deadwood and stand inventory

### 3.5.1 Deadwood

In order to measure the deadwood volume and its temporal dynamics, different survey methods were used.

SDW: Standing deadwood. During the baseline survey, all trees and stumps of at least 7 cm in diameter were recorded in the radius of 17.8 m (A = 1000 m$^2$) around the centre of the three VIP plots of each site. With this baseline assessment, many detailed parameters (condition, position, diameter, height, bark) were documented. Afterwards, every six months, the condition was re-examined.



DDW: Downed deadwood. To record the volume and condition of the lying deadwood in all its dimensions, the line intersect

sampling was used. This method is based on the USDA Forest Service forest inventory (Waddel, 2002; Woodall et al., 2019) but was adapted (Schirrmacher and Clerc, 2023) to the experimental design in the PYROPHOB project. The sampling was used on all plots (VIP and ITP) and covers all deadwood starting from a diameter of 0.1 cm: Three transects are laid out from the plot center in 30°, 150°, and 270° direction, each with a length of 7.3 m, and all intersections with deadwood are recorded.

### 3.5.2 Stand inventory data by terrestrial laserscanning

Based on annual measurement campaigns by terrestrial laserscanning (TLS), stand inventory parameters were estimated for all VIP plots. A circular area of 1000 m$^2$ (radius = 17.84 m) was covered on each plot by a total of 9 individual scans with a FARO Focus S scan terrestrial laser scanner (FARO Technologies Inc., 2021). The data set presented here includes two to three annual scan campaigns (TB: 2021-2022; JB: 2021-2023). Independent of individual tree vitality (live or dead), changes in standing trees for the main species *Pinus sylvestris* are represented by the following variables: number of standing trees,

total stem volume, basal area, tree mean height, total crown projection area. Registered and optimized point clouds on the plot level were automatically processed by script-controlled functions of the R packages TreeLS (de Conto, 2023) and lidaRtRee (Monnet, 2023) to detect and model single trees and estimate basic inventory data, stem volumes, and crown projections.

## 3.6 Vegetation

### 3.6.1 Ground vegetation data grid Treuenbrietzen 2020

This dataset provides additional vegetation data to the PYROPHOB plots for the Treuenbrietzen study area in the second year after the fire. Vegetation was surveyed between May and October 2020 on circular plots with a 10 m radius (A=314.16 m$^2$) on a systematic grid of 100 x 100 m over almost the entire burned area. On each plot, the percentage cover of the bryophyte and the herb layer was visually estimated. All vascular herbaceous plant and bryophyte species were recorded and the cover of each species was estimated using a refined Londo scale (Zacharias, 1996). The nomenclature for vascular plants (except

aggregate species) follows the Euro+Med PlantBase (Euro+Med, 2006) and the nomenclature for bryophytes follows the European checklist (Hodgetts and Lockhart, 2020). Species were assigned to regeneration strategies (invader, seed banker, sprouter, bryophyt, see Schüle et al., 2025a). We assessed post-fire forest management, fire severity, pre-fire stand age and pre-fire forest continuity for each plot (Schüle et al., 2023, 2025a) and provide the exact spatial location of each plot. This dataset has been analysed in a separate manuscript to explore the effects of pre-fire land use legacies and post-fire forest management

on vegetation and is currently under review Schüle et al. (2025a).

### 3.6.2 Vegetation data of the PYROPHOB plots 2020-2024

Vegetation was recorded on all ten plots per site in each year (2020-2024) where possible. However, due to ammunition and the second fire in 2022, not all sites could be surveyed every year. Tree, herb, bryophyte and lichen species were surveyed on 10 m radius circular plots (area = 314.16 m$^2$) between May and August on the burned sites and between July and October on

the unburned control sites. The cover of all species was estimated using a refined Londo scale (Zacharias, 1996). Artificial tree regeneration, which was planted or sown after the fire, was excluded from the dataset. The nomenclature follows the Euro+Med PlantBase (Euro+Med, 2006) for vascular plants (except aggregate species), the European checklist of bryophytes (Hodgetts and Lockhart, 2020) and the German Red List for lichens (Wirth et al., 2011). The species were categorized according to a regeneration strategy (similar to that of Schüle et al. (2025a)), status (native, archaeophyte, or neophyte) and a growth type,
based on information regarding clonal reproduction and lifespan (Klotz et al., 2002).

### 3.6.3   Tree rejuvenation

In order to record tree regeneration as a key indicator of the success of forestry treatments, the emergence of tree regeneration, as well as the growth of sown and planted trees, was systematically investigated. Two different methods were used for this.

Tree rejuvenation at VIPs: All trees up to a DBH (diameter breast height) of 7 cm within a radius of 10 m (A = 314.16 m$^2$)
were documented once a year after the vegetation period in terms of height and root collar diameter. The type of regeneration was classified into "naturally established", "planted" or "sown". Tree status was differentiated into "alive", "cut" (one of the plots served as a compensation area planted with sessile oaks, so natural regeneration was cut back to protect and promote the planted oak), "bent over" or "dead". Damages of the leading shoot were classified in the categories "browsed", "broken" or "else" (e.g. desiccated, fungi). Leaf damage was recorded when more than 30 % of the leaves showed significant damage.
Tree rejuvenation at ITPs: In order to be able to monitor individual tree growth as well, each tree was marked with an individual number within a radius of 2 m (A = 12.57 m$^2$). All trees taller than 10 cm in height and smaller than 7 cm in DBH were surveyed every winter. Except for leaf damage, the variables recorded are the same as for the VIP survey. Trees with a height < 10 cm were only identified and counted.

### 3.6.4   Canopy cover

Canopy cover was estimated using a spherical crown densiometer with a convex mirror following the methodology described by Lemmon (1956). The observations were made during the growing seasons of 2020 and 2021 on all VIP plots.

### 3.7   Fauna

### 3.7.1   Xylobiontic Beetles

The surveys of the saproxylic beetle fauna were conducted between 2021 and 2023 at a total of 16 study sites, with 11 located
in TB and 5 in JB. Sampling was carried out annually and continuously from mid-April to the end of August. In 2022, sampling on seven plots in TB had to be terminated early in June due to another wildfire event. In the following year (2023), only two of the originally seven fire-affected plots could be resampled. Saproxylic beetles were sampled using flight-interception traps. At each site, two different trap types were placed near two randomly selected VIP plots. First, window traps consisted of a crossed pair of transparent plastic shields (28 x 43 cm) connected to a funnel and a collection jar. These traps were mounted
directly onto the trunks of standing dead pine trees at a height of approximately 3.5 m (measured to the upper edge). Second,





Lindgren funnel traps (Lindgren (1983), 11-unit funnel trap, 100 x 19 cm) were suspended on ropes between two dead pine trees at a comparable height of around 3.5 m. Additionally, a single canopy funnel trap was deployed in each region. In TB, canopy sampling was conducted only in 2021, as no suitable standing pine trees remained from 2022 onward. In contrast, canopy sampling in JB was performed throughout all three years of the study. Salt solution was used as the sampling fluid in the collection jars of both trap types. All captured saproxylic beetles were identified at the species level, with the exception of members of the family Staphylinidae (rove beetles).

### 3.7.2 Moths

The moth abundance was sampled between mid-March and early November (2021-2023) on the VIP plots dependent on weather conditions and lunar cycle (Heinken et al., 2024). All three VIP-plots were instrumented with an automated window cross light trap. Each site was sampled five to ten times per year, with reduced sampling for the TB sites after the second fire. To sample the moths, automated window cross light traps were used. The trap was powered by power banks with a duration of up to six hours per night after sunset. Afterwards, each individual was identified to species level after Steiner (2014) and measured in dry weight and wing index. Species were assigned to ecological and functional traits (such as habitat preferences, caterpillar food plant or distribution), as well as red list status (Gelbrecht et al.). The data included here only include the raw data with the species identification.

### 3.7.3 Mammals

As a wildfire is a disturbance in the vegetation structure, it is also a loss or change of the habitat situation of animals. To monitor the impact and reestablishment of mammals at each site, two locations were selected and identified as positions 1 and 2 to improve coverage in very heterogeneous sites. Subsequently, a camera trap was installed at position 1 for a duration of 15 days, following which it was relocated to position 2 for an additional 15 days. The locations are independent of plot positions, and available in a separate file. Surveys were conducted four times a year to study possible seasonal changes in mammal behaviour. Seasons were defined as winter from December to February, spring from March to May, summer from June to August, and autumn from September to November. The resulting approx. 140,000 digital images were processed visually for sightings of the targeted mammal species.

## 3.8 Remote sensing and ground-based imaging

### 3.8.1 UAV imagery

To gain a deeper understanding of spatial relationships, field-collected data on vegetation and forest structure were supplemented with remote sensing. For TB, unmanned aerial vehicles (UAV, i.e. Mavic Pro, DJI, China) were utilized to capture RGB imagery with sub-decimeter ground resolution. The image acquisition took place once a year at the end of spring from 2019 to 2023. The strip-wise flight patterns allowed for sufficient overlap (longitudinal: 75-90%, transversal: 68-81%) when flown at an altitude of 120 m. A workflow for image mosaicking was created following USGS guidelines (United States Ge-



ological Survey, 2017) and customized for the specific site requirements. As the orthomosaics were created with a focus on image quality, an improvement in the surface model creation may be recommendable for the investigation of structures on the sub-meter scale like ploughing ridges, shrubs, etc. Therefore, we share the raw images from each flight. The mosaicked orthophotos are available at deadtrees.earth, an open-access platform that hosts a global database of high-resolution aerial imagery with labeled (i.e. manually classified) standing deadwood. This extensive dataset serves as a foundation for understanding tree mortality patterns worldwide.

### 3.8.2 Burn severity

The Burn severity was evaluated using the Difference Normalized Burn Ratio Index (dNBR; Key and Benson (2005)) for both study areas, utilizing multispectral satellite data (Sentinel-2 with 10–20 m spatial resolution; 2018-08-19 for TB, and 2019-06-23 for JB). This index was also used for the primary delineation of the study sites. Since this spatial scale offers limited information at the plot level, two additional indices were used, relying on bands with an increased resolution of 10 m. Using the NIR/green ratio (Daughtry et al., 2000) in JB and the MCARI1 (Haboudane et al., 2004) in TB allowed classifying the data into "burn class" categories, distinguishing between unburned, surface fire, and crown fire areas. The indices were chosen according to the best visual correspondence to the mentioned classes. The respective multispectral raw data can be acquired for free from the stated resources; here we only included the above mentioned derived indices as geospatial data. As ground truth data for the burn severity, the border of the burned area in TB was recorded in the field between autumn 2021 and spring 2022. This border was mostly clearly visible from the consumed humus and bryophyte layers as well as charred stems inside the burned area. The mapped area is provided as an ESRI-shapefile in the dataset.

### 3.8.3 Photo monitoring

Photo monitoring served for documenting the forest regrowth throughout the project period. For that purpose, photographs were taken at approximately three-month intervals using full-frame mirrorless cameras Z5, Z6, and D780 (Nikon, Japan) for one selected plot of each site. For scale, a survey rod with 50-cm-intervals is included in the photos.

Additionally, photos of the instrument setup are included in a separate folder.

Due to high resolution data, file size is around 20 MB for each image. Therefore we also provide thumbnails of the images, with file sizes of roughly 100 kB each.

### 3.9 Synthesis data set

To study the effects of post-fire forest management on several ecosystem components, data collected in 2021 (and late 2020 or early 2022) in the TB study area were analysed together. We decided to use this reduced time span for the synthesis data set, as it allows the largest number of study sites to be included. For this purpose, the above-mentioned specific datasets needed to be converted to the same spatiotemporal resolution. This joint dataset comprises mineral topsoil properties (pH, nitrogen content, phosphorus, exchangeable magnesium, potassium, and calcium), volume of lying deadwood, herb layer cover and



total biomass (above and below ground) of tree regeneration (natural and artificial), herb and bryophyte layer allometrically estimated with Schüle and Heinken (2024), species richness of vegetation (vascular plants and bryophytes) and saproxylic beetles. The continuously recorded variables air temperature, relative humidity, soil temperature, and soil moisture were used to produce aggregated indicators for specific purposes (heat stress, drought stress, maximum temperatures, vapour pressure deficit (VPD)) and were aggregated over time to a single value for each plot. Where a variable was not assessed on all plots of a site (e.g. only on the VIPs), the mean values of the assessed plots of each site were used to replace missing values in other plots of the same site in the dataset.

## 4 Relevant data already published or provided by third parties

The following subsections highlight relevant data sets that have been published already or are provided by other organizations or channels, but that we consider to be potentially helpful to users of the data presented in section 3.

### 4.1 Landuse and DEM

The state agency of Brandenburg provides various geoinformation services, among them a high-resolution digital elevation model (DEM) with a 1-meter horizontal resolution, accessible through the **Geobasis Brandenburg** service via the following WMS link: https://isk.geobasis-bb.de/mapproxy/dgm/service/wms.

### 4.2 Land cover

Geobasis Brandenburg also offers orthophotos covering the entire state, captured over a three-year period starting in 2009. These orthophotos have a ground resolution of 20 cm per pixel and include both RGB and infrared (CIR) imagery. They are particularly useful in vegetation-related studies. Older orthophotos are also available and may enable long-term investigations, though they feature lower resolutions and limited spectral coverage.

### 4.3 Soil and geohydrological maps

The **Geoportal Brandenburg** at https://geoportal.brandenburg.de/de/cms/portal/geodaten/themenkarten provides interactive maps and WMS services on soil maps in various resolutions. Various geohydrological maps (e.g. groundwater catchments, depth to the groundwater table, etc.) and station observations can be accessed via the **APW Brandenburg** portal at https://apw.brandenburg.de/.

### 4.4 Meteo data

The closest meteostations of the German Weather Service DWD are Felgentreu (ID: 1350) and Treuenbrietzen (ID: 5092). Their data can be accessed via DWD's web portal https://opendata.dwd.de/climate_environment/CDC/observations_germany/climate/.





## 5    Previous and potential use of the data

During the five years of monitoring, some of the resultant data have already been analysed. This section aims to give an overview of these studies, provide references, and discuss the potential for addressing pending questions.

In the following, we present the previously published studies based on the data presented in this paper.

### 440    5.1    Climate data

Blumroeder et al. (2022) assessed the effect of different forest management practices, using - among others - the microclimate data of the first monitoring years. They found a buffering effect of deadwood on the local temperature and water pressure deficit. It would be interesting to correlate the different deadwood dimensions (fine and coarse) with the climate data more detailed in order to be able to quantify the amount or type of deadwood at which such a buffering effect occurs. Exploiting the microclimate
data to their full length would allow the quantification of these effects also for the later stages of forest regrowth. Many of the other studies listed below underline the importance of climatic conditions - locally due to management and temporarily due to climate and weather variability. Thus, further mutual analyses of biotic observations and climatic parameters seem promising.

### 5.2    Soil

Jouy et al. (2025) incorporated the soil data (i.e. nutrients and texture) in their analysis to assess recolonization by aspen trees,
and attributed the observed high variability to a complex interaction of multiple further factors. Gerwin et al. (2025, submitted) found that soil data also showed almost complete loss of formerly thick humus layers and an increase in base saturation. Based on the soil data, they also concluded that soil conditions significantly differ between intensively managed fire sites (complete salvage logging and ploughing) and unmanaged fire sites. The data on leaching is under current analysis, regarding the relationship between DOC and the prior fire event and forest management. Likewise, the relationship between soil respiration and
the intensity of the prior fire event could be of interest. Dobkowitz et al. (2025) used the data of soil moisture and temperature to assess drought and temperature stress for the different sites. Soil moisture time series served in the calibration of a soil hydraulic model, which also used the recorded meteo-data as an input variable. The model proved to perform unsatisfactorily, so a refined approach with a better representation of root water uptake and the specific soil hydraulic properties of the sandy soils could be envisioned.

### 460    5.3    Deadwood and stand structure

The large mounts in deadwood inventories and difficult access to some of the plots required the development of an efficient and robust methodology, which was documented by Clerc and Schirrmacher (2023). The concomitant data from both conventional and laserscanning-based methods (Clerc et al., 2023) allows for the systematic comparison of both approaches. Moreover, the added geometric detail of laserscanning-based data enables a deeper analysis of deadwood effects on microclimate, species
diversity and abundance of insects. In conjunction with soil and seepage data, the role of deadwood decomposition on nutrient





fluxes could be assessed. Likewise, an in-depth analysis of the protective effects of deadwood against damage through browsing or abundance of browsers could provide valuable insights on this relationship.

## 5.4 Vegetation

The collected vegetation data demonstrate the profound changes in vegetation composition following a fire. The data also
show the rapid shifts in plant species composition in the first post-fire years, and the effects of post-fire forest management on vegetation. Schüle et al. (2023) and Blumroeder et al. (2022) conclude that natural regrowth resulting from limited interventions is strong, and emphasise the importance of potential seed trees and species that regenerate quickly, such as *Populus*, *Betula* and *Robinia*. Based on the vegetation surveys, Schüle and Heinken (2024) demonstrate the importance of pioneer species as they enhance quick biomass recovery, soil stabilization, and mitigation of mineral leaching. Jouy et al. (2025) emphasize the
high variability of regeneration and the resulting future unpredictability due to the ongoing climate change and more regularly expected extreme weather events. However, a closer examination of regrowth dynamics in relation to the prevailing observed meteorological conditions is still pending.

Allometric models were developed to estimate the aboveground and belowground plant biomass in burned and unburned Scots pine forests (Schüle and Heinken, 2024). The above- and belowground biomass of various herb, grass and tree species,
as well as aboveground biomass of several bryophyte species, was collected in 2021 across all study sites in Treuenbrietzen and Jüterbog. A detailed description of the sampling can be found in (Schüle and Heinken, 2024). The data of the collected biomass samples, which were used to build the allometric models, were published as online supplementary material of Schüle and Heinken (2024). They can be further used to assess post-fire carbon pools. Additional open scientific questions remain concerning the effects of land use legacy, fire severity, and spatial factors on post-fire vegetation.

## 485 5.5 Remote sensing and ground-based imaging

The acquired UAV-imagery helped in the development of an automated identification of standing deadwood (Schmehl, 2025). They were included in an intercomparison project for developing detection methods for tree mortality (Mosig et al., 2024). In a MSc thesis, the potential of multispectral satellite imagery for identifying habitat types could be shown. In conjunction with the detailed vegetation surveys, the UAV-data offer a rich test field for future tests on their usability for advanced image
classification methods.

The repeated photo-shots of selected plots have served as an effective way to illustrate the observed regrowth, especially to the non-scientific public. Thanks to their high resolution, they may also hold the potential for the estimation of unmeasured parameters (e.g. combustible litter inventory) over time.

## 5.6 Synthesis

For an integrative assessment of the data, several analyses have been undertaken on the synthesis data (see Sect. 3.9), employing multivariate techniques such as PCA and mixed effects models Schüle et al. (2025b). So far, the published studies focus mainly





on ecological questions that compare different types of management and investigate how biotic and abiotic factors such as soil properties, fire severity, browsing, or potential seed trees influence the ecological outcome. Until now, only two of them (i.e. Schüle et al., 2023; Jouy et al., 2025) include impacts by fauna, but focus on the relation between abiotic, vegetation, and vegetation structure factors.

Due to the lack of replication in several treatments and data gaps caused by technical issues and the second fire in TB, a comprehensive comparison of all factors across all plots and the entire time period is not fully possible. However, the data present numerous opportunities for conducting more localized analyses (as mentioned in the previous subsections), like examining temporal changes at specific sites, which may not always include all datasets or plots.

Additional to the above-mentioned studies, the project partners supervised a total of more than 47 student theses (BSc, MSc), each addressing specific research questions. These theses have been submitted to hosting universities of the respective working groups, i.e. Brandenburg University of Technology Cottbus-Senftenberg, Eberswalde University for Sustainable Development, and University of Potsdam.

## 6 Conclusions

The PYROPHOB research project investigates the impacts of post-fire management strategies on forest recovery in Scots pine stands in Brandenburg, Germany. Its findings show that both pre-existing site conditions — such as soil texture, vegetation structure, and landscape context — and post-fire silvicultural treatments jointly determine forest regeneration outcomes. Unmanaged sites often exhibited more favorable microclimatic conditions and, partially, higher regeneration of pioneer species like Populus tremula, particularly where surrounding seed sources were available and soils retained some fertility.

The project highlights the need to consider the interplay between site-specific ecological conditions, historical land use legacies, and management intensity when planning forest restoration. In particular, natural succession involving fast-regenerating pioneer species contributed significantly to biomass recovery and microclimatic buffering, even under challenging post-fire conditions. These insights underscore the importance of context-sensitive, adaptive strategies for increasing forest resilience in the face of climate change.

This data paper describes the majority of the data acquired during the project and archives them to enable future re-use and reproducibility. The value of the data set lies in its role as an interdisciplinary effort carried out soon after the wildfire, offering a 5-year snapshot for assessing forest regrowth. Evidently, especially for forest ecosystems, long-term or repeated studies and monitoring would be advantageous, especially under transient climatic conditions. In that context, the occurrence of the second wildfire presents both severe constraints but also extra opportunities in the data.

Despite its limitations, this data set lays the groundwork for such extensive analyses. For similar future studies we recommend a well-planned study design, balancing the requirements and restrictions of the different disciplines while maximizing coherence in terms of spatiotemporal overlap in scale and representativity. Likewise, a platform for a low-threshold data interchange, including well-defined internal conventions, greatly fosters interdisciplinary data use. These prerequisites ensure





that hypotheses can be tested rigorously on the appropriate spatial and temporal entities with maximum synergies between the
different disciplines.

## 7  Data availability

The published data set is organized along instruments and observed variables, and follows the structure of Sect. 3 of this
paper (Tab. 3). Each subset of data is documented in a dedicated meta-data file in "json" format. Format conventions follow
Fersch et al. (2020) and Heistermann et al. (2022) and are summarized in the file 'readme.txt' at the top level. We used EUDAT
infrastructure (https://eudat.eu), namely the services B2SHARE and B2HANDLE, in order to manage identifiers and guarantee
long-term persistence.

The described data are distributed over three repositories:

1. Main repository (Schmehl et al. (2025a), DOI: 10.23728/b2share.5821b57eae3b45619c2263205b1c9815): This repository contains the main part of the described data

2. Embargoed repository (Schmehl et al. (2025b), DOI: 10.23728/b2share.54a4388659c94a518f0bc39fd191cb64): As some of the described dataset are currently under analysis in ongoing studies, their contributors requested them to become available only after 31. March 2026. Access opens automatically after this date. The embargo concerns some of data on the meteo-observations, soil respiration, the fauna and additional years to the vegetation surveys.

3. Imagery data (Schmehl et al. (2025c, d), DOIs: 10.23728/b2share.d49631857d1345c48be9e36b62214e6d and
10.23728/b2share.51bdf4b6dc854873b6ff44fdddbf4c3b):

   As the remote sensing imagery is especially extensive, we used two separate repositories to allow for a more granular data access that allows specifically downloading parts of interest instead of excessively large chunks.

Table 3 summarizes the allocation of each data subset to the repositories and its embargo state. Besides the actual data, each
folder contains a JSON file (readable ASCII text) holding metadata and additional format descriptions, where these differ from
the conventions described in Sect. 3.1.



**Table 3.** Organisation of the data subsets in the data repositories. See also Table 2 for corresponding details on the data subsets.
Repository keys: 1: Schmehl et al. (2025a), 2: Schmehl et al. (2025b); 3.1: Schmehl et al. (2025c); 3.2: Schmehl et al. (2025d)

| Sect. | Subset | Repository | Path | Publication Type |
|---|---|---|---|---|
| 3.3.1 | Geo data | 1 | geodata.zip | open |
| 3.3.1 | Meteo data | 2 | climate_weatherstation.zip | embargo |
| 3.3.2 | Precipitation | 1 | climate_precipitation.zip | open |
| 3.3.3 | Microclimate | 2 | climate_microclimate.zip | embargo |
| 3.4.1 | Humus and soil properties | 1 | soil_conditions.zip soil_humus.zip | open |
| 3.4.2 | Litter decomposition | 1 | soil_litter_decomp.zip | open |
| 3.4.3 | N mineralization | 1 | soil_N_mineral.zip | open |
| 3.4.4 | Soil solution | 1 | soil_solution_lys.zip | open |
| | | 2 | soil_solution_succup.zip | embargo |
| 3.4.5 | Soil respiration | 2 | soil_respiration.zip | embargo |
| 3.4.6 | Soil moisture and soil temperature time series | 1 | soil_moisture_timeseries.zip soil_temperature.zip | open |
| 3.4.6 | Soil moisture, areal sampling | 1 | soil_moisture_campaigns.zip | open |
| 3.5.1 | Deadwood inventory | 1 | deadwood.zip | open |
| 3.5.2 | stand structure | 1 | stand_structure.zip | open |
| 3.6.1 | Ground vegetation (grid) | 1 | vegetation_species_grid.zip | open |
| 3.6.2 | Ground vegetation (plots ) | 2 | vegetation_species_plots.zip | embargo |
| 3.6.3 | Tree rejuvenation | 1 | vegetation_rejuvenation.zip | open |
| 3.6.4 | Canopy cover | 1 | vegetation_canopycover.zip | open |
| 3.7.1 | Xylobiontic Beetles | 2 | fauna_xylo_beetles.zip | embargo |
| 3.7.2 | Moths | 2 | fauna_moths.zip | embargo |
| 3.7.3 | Mammals | 2 | fauna_mammals.zip | embargo |
| 3.8.1 | UAV imagery | 3.1, 3.2 | UAV_[YYYYMMDD].zip | open |
| 3.8.2 | Burn indices | 3.1 | burn_indices_burn_class.zip burn_indices_dNBR.zip | open |
| 3.8.3 | Photo monitoring | 1 | photomonitoring_plots.zip photomonitoring_intruments.zip photomonitoring_thumbnails.zip | open |



*Author contributions.* MTS and TF drafted the manuscript, coordinated the related data management and homogenization and led the writing of the manuscript; PI designed the layout of the study sites together with other members of the consortium and coordinated the project. The various co-authors acquired, processed and contributed the following datasets: YA and MSü with the help of all participating authors (synthesis data), AB, DC, PLI, LR, MSi and JS (tree rejuvenation and deadwood), CB (moths), SD and TF (soil moisture, precipitation), WG, TR and FR (soil), HH (beetles), TH and MSü (vegetation data), CH and JS (stand structure), KG, FJ and AVH (soil respiration and solution), LL (mammals), FR (soil solution), SR and PLI (meteo and microclimate). All authors contributed to writing and proofreading the manuscript.

*Competing interests.* The authors declare no competing interests.

*Acknowledgements.* The PYROPHOB project was funded by the Forest Climate Fund (Waldklimafonds) via the Fachagentur Nachwachsende Rohstoffe e.V. (FNR) by the Federal Ministry of Food and Agriculture (Bundes- ministerium für Ernährung und Landwirtschaft, BMEL) and the Federal Ministry for the Environment, Nature Conservation and Nuclear Safety (Bundesministerium für Umwelt, Naturschutz und nukleare Sicherheit, BMU) (grants no: 2219WK50A4; 2219WK50B4; 2219WK50C4; 2219WK50D4; 2219WK50E4; 2219WK50F4; 2219WK50G4; 2219WK50H4). The project's initiation was only made possible by the invitation of the municipal forester Dietrich Henke. Jeanette Blumröder played a vital role during the planning, application, and implementation of the project, and the deployment of the microclimatic instrumentation. We express our gratitude to Dr. Tilo Geisel of www.naturfoto-geisel.com for conducting the photo monitoring. We also thank our field technician Silvio Vogt (BTU) as well as numerous BSc and MSc candidates and student assistants for their contributions. We are indebted to the landowners (especially the town of Treuenbrietzen and the Waldgenossenschaft Bardenitz eG), the responsible foresters, and the authorities for their support. We thank the responsible state environmental offices of Brandenburg for the field permit (according to §67 BNatSchG) and the special permit for the capture of animals (according to §45 BNatSchG). We are grateful to EUDAT for providing a powerful infrastructure for data storage and handling.

AI tools were used for style and spell checking during the writing of the manuscript.





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
