# Peer review of "Monitoring abiotic and biotic parameters of forest regrowth under different management regimes on former wildfire sites in northeastern Germany – data from the PYROPHOB project"

_Earth System Science Data, 2025_

## Author Comment (AC1)

**Author Response to Referee #1**

**Monitoring abiotic and biotic parameters of forest regrowth under different management regimes on former wildfire sites in northeastern Germany – data from the PYROPHOB project**

Marie-Therese Schmehl et al.
*Earth Syst. Sci. Data Discuss.,* `doi:10.5194/essd-2025-313`

RC: *Referee Comment*,     AR: *Author Response*

Dear Referee,

thank you very much for your positive response, and for taking the time and effort to examine the manuscript and the data set.

Your comments, even if minor, are very helpful to clarify certain aspects in the manuscript. Especially the detailed revision of the data set improves its quality. Please find a point-by-point reply below.

Kind regards,
Marie-Therese Schmehl (on behalf of the author team)

**Comments and responses**

RC:   *On P.1, L.20: "Among the above-mentioned hazards..."*

It would be good mentioning that all these drivers of forest stress (listed on L.15) are more or less interlinked. For example, climate change increases intensity and frequency of wildfires.

Also, in the list on L.15, natural disasters are explicitly listed. Isn't a wildfire also a natural disaster?

AR:   Of cause the drivers of forest stress can be interlinked, though in different extents and wild fires are one example to natural desasters. We will adapt the sentence to clarify the above mentioned issues.

RC:   *On P.22, L.505: The authors mention 47 student theses that have been written within this project. Are these publicly available? Would it be worthwhile to compile them and upload them as supplementary data with a unique DOI? Otherwise, I would omit mentioning them in this data paper.*

AR:   We appreciate this suggestion, but regulations at the involved insitutions differ widely and the respective theses are not fully publically available due to copyright issues. Thus, we will delete the sentence.

RC:   *There are three data sets associated with this paper. The data in general seems of high quality. I have*

*commented some minor issues I had when I reviewed them.*

AR: We appreciate your time and effort revising each one of them. Below we answer to the ones commented different than "good" (they all belong to the main repository)

**RC:** *5. Photomonitoring instruments [ok, more information on individual pictures might be helpful]*

AR: We will name the files acording to their date and the english instrument name.

**RC:** *6. Photomonitoring thumbnails [site pictures are good, same comment applies for pictures of instruments]*

AR: Same as above.

**RC:** *8. Soil condition data [good, dates are not in order]*

AR: We can see your point in ordering all data according to their date, as the data set consists of time series. As the soil condition was investigated only once we found it more practical to order the data according to their plot name (same as soil humus). We formatted all dates the same way, so in case of machine based reading of the data it can be easily rearranged if the dates are preferred for ordering.

**RC:** *10. Soil N mineralisation data [see below]*

What does "Date_0" and "Date_Exp" mean? It's neither explained in the json file, nor in the article. Seems to be start and end dates?

What does "L" mean in the variable "depth"? Only S and M are explained.

AR: The interpretation of "Date_0" and "Date_Exp" ist correct. We will add a respective sentence for clarification.

**RC:** *11. Soil moisture data [see below]*

This might be me, but I don't understand exactly what (0-5|A-F) means. Could you please explain?

Column names in epsilon_series are not explained.

AR: Indeed the suffix (0-5|A-F) in the column names are not intuitive to understand. They refer to the depths of the doubled profiles used for these measurements. We will clarify with an adapted description of the "Technical Details" section.

Same goes for the suffixes in the column names in the epsilon series, where - as you note correctly - the actual variable description is missing. We will add it.

**RC:** *13. Soil moisture campaigns data [ok, columns in voltage_epsilon_theta_FDR.txt could be explained a bit more]*

AR: We will add a sentence and refer to the (new) description of the epsilon_series.

---

## Author Comment (AC2)

**Author Response to Referee #2**

**Monitoring abiotic and biotic parameters of forest regrowth under different management regimes on former wildfire sites in northeastern Germany – data from the PYROPHOB project**

Marie-Therese Schmehl et al.

*Earth Syst. Sci. Data Discuss.,* `doi:10.5194/essd-2025-313`
* * *
RC: *Referee Comment*,     AR: *Author Response*

Dear referee,

thank you very much for your positive response, and for the time and effort spent to examine the manuscript and the data set.

Your comments, even if minor, are very helpful. Please find a point-by-point reply below.

Kind regards,
Marie-Therese Schmehl (on behalf of the author team)

**Comments and responses**

**RC:** *General comments: In the Abstract and introduction, the authors post the word "holistic monitoring" several times. It would be interesting, if it could be clarified at somewhere what holistic what mean in this context. From a reader and researchers' perspective, it would be extremely good to know for which aspects/analysis the dataset has the highest potential and where it has its weaknesses or where additional data would be needed. (edit: I see that this appears somewhat under 1.3 "Structure" – I think it would fit better to the introduction).*

**AR:** We will supplement the respective sentences to point out our notion of "holistic". We can see the point and understand your intuition for outlining the importance of the most promising aspects. The respective reference in section 1.3 was misleadingly pointing only to the Conclusions section. We will fix the respective reference to guide the reader also to section "Previous and potential use of the data", explicitly addressing the various potential of the dataset. We suggest to leave the aspect on the potential and the weaknesses of the dataset to this concluding sections, as it largely refers to the presented data and requires their presentation beforehand.

**RC:** *One question coming to my mind is: How was the forest structure before burning? Was it homogeneous?*

**AR:** We will add respective information on the age of the affective forest stands in the description, accompanied by a reference to table 2. Further detailed information on stand properties before the fire is unavailable, as research only started after the event.

**RC:** *Specific comments: L. 6-7: UAV-based remote sensing, and photo monitoring: please provide details as for the other groups of data (which parameters/type of data?)*

**AR:** remote sensing and photo monitoring served "stand structure and spatial overview" and "temporal succession". We will add that to the sentence respectivly.

**RC:** *L. 47-48: "PYROPHOB explicitly considers a wider selection of management options to assess their impact on natural regeneration and other related environmental variables." à which ones? Would make sense to list them here imo.*

**AR:** We will add a reference to Tab. 1 which list the relevant mamangement options.

**RC:** *l.101-102: the criteria for the pre-screening is unclear (e.g., similar soil type /geology for all plots, aspect, etc.?)*

**AR:** The pre-selection for the designation of the study sites aimed at identifying areal entities with reasonable homogeneity in the mentioned properties (burn intensity, soil type, relief, distance to ground water, stand age and type); and also avoiding local untypical singularities to maximize transferability. We will add a respective sentence.

**RC:** *Fig. 1: the no removal and unburnt colors are difficult to distinguish*

**AR:** We used a colorblind friendly scheme according to Crameri et. al (2020), but will try to darken the "unburnt" color, and/or change the thickness of lines to improve the distinguishability.

**RC:** *3.1: For all temporal data, it would be useful to include the temporal resolution in the table*

**AR:** We did not want to overload the table with detail information, as it can be reviewed easily in the accompanying description file of each data set. In order to preserve formatting not bloating the table we will additionally mark all data with regular and fixed temporal resolution using table footnotes.

**RC:** *l.267: in table 3.1 it states soil moisture is recorded in 30cm, here it is spatially distinct until 60cm This might be a misunderstanding. Soil moisture is continously measured down to 100cm. Additional punctual campaigns for spatial correction were conducted down to 30cm. This we will clarify by adding depth information to table 2 to better distinguish both datasets.*

**RC:** *Technical comments: not sure if it is a BE vs. AE thing, but usually there is a comma after (i.e.,) and (e.g.,)*

**AR:** We will revise the text accordingly and change to the suggested spelling.

**1. References**

Crameri, F., G.E. Shephard, and P.J. Heron (2020), The mis-use of colour in science communication, Nature Communications, 11, 5444. doi: 10.1038/s41467-020-19160-7

---

## Author Response (AR1)

**Author Response to Referees**

Monitoring abiotic and biotic parameters of forest regrowth under different management regimes on former wildfire sites in northeastern Germany – data from the PYROPHOB project

Marie-Therese Schmehl et al.

Earth Syst. Sci. Data Discuss., doi:10.5194/essd-2025-313

RC: Referee Comment, AR: Author Response, ☐ Manuscript text

Dear referees.

thank you very much for the positive responses, and for the time and effort spent to examine the manuscript and the data set.

With this letter, we provide the responses to both referee reports in one document. They basically correspond to our previous responses in the interactive discussion.

We hope that the revised versions of the manuscript and the dataset meet the standards of ESSD.

Kind regards,

Marie-Therese Schmehl (on behalf of the author team)

**1. Responses to referee #1**

RC: On P.1, L.20: "Among the above-mentioned hazards..."

It would be good mentioning that all these drivers of forest stress (listed on L.15) are more or less interlinked. For example, climate change increases intensity and frequency of wildfires.

Also, in the list on L.15, natural disasters are explicitly listed. Isn't a wildfire also a natural disaster?

- AR: Indeed some of the drivers of forest stress are interlinked, though in different extents, and wild fires are one example to natural desasters. We adapted the sentence to clarify the above mentioned issues.
- RC: On P.22, L.505: The authors mention 47 student theses that have been written within this project. Are these publicly available? Would it be worthwhile to compile them and upload them as supplementary data with a unique DOI? Otherwise, I would omit mentioning them in this data paper.
- AR: We appreciate this suggestion, but regulations at the involved insitutions differ widely and the respective theses are not fully publically available due to copyright issues. Thus, we deleted the sentence.

- RC: There are three data sets associated with this paper. The data in general seems of high quality. I have commented some minor issues I had when I reviewed them.
- AR: We appreciate your time and effort revising each one of them. Below we answer to the ones commented different than "good" (they all belong to the main repository)
- RC: 5. Photomonitoring instruments [ok, more information on individual pictures might be helpful]
- AR: We named the files according to their date and the english instrument name.
- RC: 6. Photomonitoring thumbnails [site pictures are good, same comment applies for pictures of instruments]
- AR: Same as above.
- RC: 8. Soil condition data [good, dates are not in order]
- AR: We can see your point in ordering all data according to their date, as the data set consists of time series. As the soil condition was investigated only once we found it more practical to order the data according to their plot name (same as soil humus). We formatted all dates the same way, so in case of machine based reading of the data it can be easily rearranged if the dates are preferred for ordering.
- RC: 10. Soil N mineralisation data [see below]
  - What does "Date\_0" and "Date\_Exp" mean? It's neither explained in the json file, nor in the article. Seems to be start and end dates?
  - What does "L" mean in the variable "depth"? Only S and M are explained.

**AR:**

- The interpretation of "Date\_0" and "Date\_Exp" ist correct. We added an explanation in the respective sentence for clarification.
- This is a mistake in the explanation file. S should actually be L (from german 'Streu'/engl. 'litter'). We corrected it.

**RC: 11. Soil moisture data [see below]**

- This might be me, but I don't understand exactly what (0-5|A-F) means. Could you please explain?
- Column names in epsilon\_series are not explained.

**AR:**

- Indeed the suffix (0-5|A-F) in the column names are not intuitive to understand. They refer to the depths of the doubled profiles used for these measurements. We clarified with an adapted description of the "Technical Details" section.
- Same goes for the suffixes in the column names in the epsilon series, where as you note correctly the
  actual variable description is missing. We added it.

- RC: 13. Soil moisture campaigns data [ok, columns in voltage\_epsilon\_theta\_FDR.txt could be explained a bit more]
- AR: We added a sentence and refer to the (new) description of the epsilon\_series.

**2. Response to referee #2**

- RC: General comments: In the Abstract and introduction, the authors post the word "holistic monitoring" several times. It would be interesting, if it could be clarified at somewhere what holistic what mean in this context. From a reader and researchers' perspective, it would be extremely good to know for which aspects/analysis the dataset has the highest potential and where it has its weaknesses or where additional data would be needed. (edit: I see that this appears somewhat under 1.3 "Structure" I think it would fit better to the introduction).
- AR: We supplemented the respective sentences to point out our notion of "holistic". Also we can see the point and understand your intuition for outlining the importance of the most promising aspects. The respective reference in section 1.3 was misleadingly pointing only to the Conclusions section. We fixed the respective reference to guide the reader also to section "Previous and potential use of the data", explicitly addressing the various potential of the dataset. We suggest to leave the aspect on the potential and the weaknesses of the dataset to this concluding sections, as it largely refers to the presented data and requires their presentation beforehand.
- RC: One question coming to my mind is: How was the forest structure before burning? Was it homogeneous?
- AR: We added respective information on the age of the affective forest stands in the description, accompanied by a reference to table 2. Further detailed information on stand properties before the fire is unavailable, as research only started after the event.
- RC: Specific comments: L. 6-7: UAV-based remote sensing, and photo monitoring: please provide details as for the other groups of data (which parameters/type of data?)
- AR: remote sensing and photo monitoring served "stand structure and spatial overview" and "temporal succession". We added that to the sentence respectively.
- RC: L. 47-48: "PYROPHOB explicitly considers a wider selection of management options to assess their impact on natural regeneration and other related environmental variables." à which ones? Would make sense to list them here imo.
- AR: We added a reference to Tab. 1 that lists the relevant mamangement options.
- RC: 1.101-102: the criteria for the pre-screening is unclear (e.g., similar soil type /geology for all plots, aspect, etc.?)
- AR: The pre-selection for the designation of the study sites aimed at identifying areal entities with reasonable homogeneity in the mentioned properties (burn intensity, soil type, relief, distance to ground water, stand age and type); and also avoiding local untypical singularities to maximize transferability. We added a respective sentence.
- RC: Fig. 1: the no removal and unburnt colors are difficult to distinguish
- AR: We used a colorblind friendly scheme according to Crameri et. al (2020), but darkened the "unburnt" color to

improve the distinguishability.

- RC: 3.1: For all temporal data, it would be useful to include the temporal resolution in the table
- AR: We did not want to overload the table with detail information, as it can be reviewed easily in the accompanying description file of each data set. In order to preserve formatting not bloating the table we additionally marked all data with regular and fixed temporal resolution using table footnotes.
- RC: 1.267: in table 3.1 it states soil moisture is recorded in 30cm, here it is spatially distinct until 60cm
- AR: This might be a misunderstanding. Soil moisture is continously measured down to 100cm. Additional punctual campaigns for spatial correction were conducted down to 30cm. This we clarified by adding depth information to table 2 to better distinguish both datasets.
- RC: Technical comments: not sure if it is a BE vs. AE thing, but usually there is a comma after (i.e.,) and (e.g.,)
- AR: We revised the text accordingly and changed to the suggested punctuation.

**3. Other changes not related to referee comments**

We applied a few corrections that were not motivated by any referee comment.

- In section 5.2 we slightly adapted according to the current state of the analyses.
- Due to an adaption in the synthesis data set in the corresponding publication, we adapted the data set the same way (mainly excluding the mean values of the PCA data set and merging them into one table).
- We updated the DOIs to the latest version of the datasets in the manuscript (abstract, data availability section, references).

**References**

Crameri, F., G.E. Shephard, and P.J. Heron (2020), The mis-use of colour in science communication, Nature Communications, 11, 5444. doi: 10.1038/s41467-020-19160-7